# The Relationship between Vitamin D Metabolites and Androgens in Women with Polycystic Ovary Syndrome

**DOI:** 10.3390/nu12051219

**Published:** 2020-04-26

**Authors:** Jakub Mesinovic, Helena J. Teede, Soulmaz Shorakae, Gavin W. Lambert, Elisabeth A. Lambert, Negar Naderpoor, Barbora de Courten

**Affiliations:** 1Department of Medicine, School of Clinical Sciences at Monash Health, Faculty of Medicine, Nursing and Health Sciences, Monash University, Clayton 3168, Australia; jakub.mesinovic@monash.edu; 2Monash Centre for Health Research and Implementation, School of Public Health and Preventive Medicine, Monash University, Clayton 3168, Australia; Helena.Teede@monash.edu (H.J.T.); soulmaz.shorakae@monash.edu (S.S.); negar.naderpoor@monash.edu (N.N.); 3Iverson Health Innovation Research Institute, Swinburne University of Technology, Melbourne 3000, Victoria, Australia; GLambert@swin.edu.au (G.W.L.); elisabethlambert@swin.edu.au (E.A.L.)

**Keywords:** polycystic ovary syndrome, vitamin D, vitamin D-binding protein, androgens, testosterone

## Abstract

Polycystic ovary syndrome (PCOS) is the most common endocrine disorder among women of reproductive age, with hyperandrogenism present in up to 90% of affected women. Some evidence suggests a link between vitamin D deficiency and PCOS features via insulin resistance and inflammation. Our aim was to explore the relationship between biochemical markers of vitamin D status and androgens in women with PCOS. This cross-sectional study used bio-banked samples from 46 pre-menopausal women with PCOS (mean ± SD: age 30 ± 6 years; BMI 29 ± 6 kg/m^2^). We measured 25-hydroxyvitamin D (25[OH]D), vitamin D-binding protein (DBP), total testosterone, sex hormone-binding globulin (SHBG), and calculated the free androgen index (FAI) and bioavailable and free 25(OH)D. Fasting glucose and insulin were used to calculate the homeostatic model assessment of insulin resistance (HOMA-IR) and body fat percentage was determined via dual energy x-ray absorptiometry. High-sensitivity C-reactive protein (hs-CRP) was measured as a marker of inflammation. DBP was positively associated with total 25(OH)D and expectedly, negatively associated with free 25(OH)D. There were no associations between vitamin D metabolites and total testosterone, SHBG or FAI, even after adjusting for age, body fat percentage, HOMA-IR and hs-CRP. We found no associations between vitamin D metabolites and androgens in women with PCOS. Studies that have identified a vitamin D–androgen link have largely relied on methodology with numerous pitfalls; future studies should exclusively use gold-standard measures to confirm these findings in this population.

## 1. Introduction

Vitamin D deficiency is a major public health problem that affects 20% of Australian women [1,2,3]. Vitamin D deficiency has been implicated in the development of polycystic ovary syndrome (PCOS) [4]. PCOS affects approximately one in five women of reproductive age and is associated with insulin resistance and hyperandrogenism [5], which might be influenced by components of vitamin D metabolism.

Vitamin D receptor (VDR) mRNA is expressed in several reproductive tissues in women, including the ovaries [6]. Ovaries synthesise estrogens and androgens, and the active form of vitamin D, 1,25 dihydroxyvitamin D, increases estradiol and progesterone, but does not appear to affect androgen synthesis in human ovarian cells [6]. Despite this, studies in women with PCOS have reported negative associations between vitamin D and androgen levels [7,8], and vitamin D supplementation can normalise androgen levels in this population [9,10]. Meta-analyses have reported inconsistent findings with respect to the effect of vitamin D supplementation on androgens in women with PCOS [11,12].

To the best of our knowledge, no studies have investigated the relationship between androgens and other key components of vitamin D metabolism, such as vitamin D-binding protein (DBP), in women with PCOS. DBP regulates free and total 25-hydroxyvitamin D (25[OH]D) concentrations and is involved in transporting vitamin D metabolites into the kidneys, and likely the parathyroid gland and placenta [13]. According to the free hormone hypothesis [14], only free 25(OH)D is biologically active and thus, disparities in DBP concentrations in previous studies may have contributed to the convoluted relationship between vitamin D and androgen metabolism.

In a previous cross-sectional study, we reported that total and free 25(OH)D concentrations were similar in women with and without PCOS, but DBP was lower in women with PCOS [15]. However, in that study, we were unable to investigate the relationship between vitamin D and androgen metabolism, as androgens were measured using different assays. Since then, we used a subset of this cohort that had testosterone levels measured using high-performance liquid chromatography with tandem mass spectrometry to explore the relationship between vitamin D and androgens in women with PCOS. We hypothesized that free 25(OH)D would be inversely associated with testosterone levels.

## 2. Materials and Methods

### 2.1. Study Design and Setting

This is a cross-sectional study of bio-banked plasma samples from an anthropometrically and biochemically well-characterised population of 46 pre-menopausal women with PCOS aged between 20 to 45 years (NCT01504321) [16]. Women were recruited prospectively through community advertisements in Melbourne, Australia, and participated between January 2013 and March 2015. Baseline serum samples were bio-banked and stored in aliquots at −80 °C. Participants were primarily Caucasian (*n* = 36; 78%). PCOS was diagnosed by an endocrinologist using the Rotterdam criteria [17], with the presence of two of the following three components: oligo/anovulation (or cycle length <35 days), clinical (hirsutism) or biochemical hyperandrogenism, and polycystic ovarian morphology on ultrasound (presence of 12 or more follicles measuring 2–9 mms in one or both ovaries) [18]. Hirsutism was evaluated using a modified Ferriman–Gallwey scoring (m-FG score) system [19] and was defined as an m-FG score above or equal to 8 in Caucasian and above or equal to 6 in Asian women. Vitamin D status was determined via Institute of Medicine cut-points [20]: 25(OH)D concentrations ≥50nmol/L were considered sufficient, 25(OH)D <50nmol/L was considered insufficient, and 25(OH)D <30nmol/L was considered deficient. Participants were not on metformin for at least 1 month nor on any contraceptive pills for at least 3 months prior to data collection. Participants taking vitamin D or multivitamin supplements based on self-reported medication lists were excluded. This study was approved by the Monash Health and the Alfred Hospital Human Research Ethics Committees, and all participants provided written informed consent.

### 2.2. Anthropometric and Body Composition

All participants had an initial screening appointment where weight and height were measured. Body mass index (BMI) was calculated using weight in kilograms divided by height in meters squared (kg/m^2^). Body composition was examined in all women using total body dual-energy X-ray absorptiometry (DXA) (Lunar Radiation Corp., Madison, WI, USA).

### 2.3. Blood Biochemistry

Fasting blood samples were collected via venepuncture. Serum total 25(OH)D concentration was determined using the direct competitive chemiluminescent immunoassay (CLIA) method on a LIAISON analyser (DiaSorin Inc., Stillwater, MN, USA) with intra- and inter-assay coefficients of variations (CV) of < 4% and < 10%, respectively. Serum DBP was measured in duplicate using a polyclonal competitive ELISA assay (Abcam, AB108853) with intra-assay CV of 4.9% and inter-assay CV of 7.3%. Serum albumin was measured using an automated colorimetric method carried out on a Beckman Coulter AU5812^®^ System. We calculated free 25(OH)D and bioavailable 25(OH)D using equations provided by Powe and colleagues [21]. The affinity binding constants for 25(OH)D with DBP and albumin were 7 × 108 M^−1^ and 6 × 105 M^−1^, respectively, which was determined by Bikle and colleagues by centrifugal ultrafiltration dialysis [22]. Total testosterone was measured using high-performance liquid chromatography with a tandem mass spectrometry method using a liquid sample extraction (AB Sciex Triple Quad 5500 LC/MS/MS system; Mt Waverley, VIC, Australia). The Access sex hormone-binding globulin (SHBG) assay was performed using a sequential two-step immunoenzymatic (“sandwich”) assay (A48617) carried out on a Beckman Coulter Unicel DXI 800 (Beckman Coulter, Lane Cove, NSW, Australia). The free androgen index (FAI) was calculated as (total testosterone × 100)/SHBG. Serum fasting glucose was measured using a commercial enzymatic kit (472500, Beckman Coulter) and insulin was measured using a radioimmunoassay (EZHI14-K, Linco, St. Charles, MO, USA). Homeostatic model assessment of insulin resistance (HOMA-IR) was calculated using the following formula: fasting insulin concentration × fasting glucose concentration / 22.5. High-sensitivity C-reactive protein (hs-CRP) was measured by immunoturbidimetric assay (reference range <= 5 mg/L).

### 2.4. Statistical Analyses

The distribution of variables was assessed using histograms and Shapiro–Wilk tests. Outcome measures were reported as mean ± SD and median (IQR) for continuous normally and non-normally distributed variables, respectively. Relationships between vitamin D metabolites and binding protein were assessed using Spearman’s correlations. Pearson’s correlations and multivariable linear regression examined the relationship between components of vitamin D and androgen metabolism. Linear regression models were adjusted for age, body fat percentage, HOMA-IR and hs-CRP. A two-tailed p-value < 0.05 was considered statistically significant. SPSS version 25 was used to perform statistical analyses.

## 3. Results

The mean age of the cohort was 29.9 ± 5.6 years. Fifty percent of these women had insufficient vitamin D concentrations and seven percent were deficient. Eighty-seven percent of these women had hirsutism measured via m-FG scores. Descriptive characteristics are presented in Table 1.

Spearman’s correlations between markers of vitamin D status are presented in Table 2. DBP was positively associated with total 25(OH)D and negatively associated with free 25(OH)D. There were no associations between DBP and bioavailable 25(OH)D. Pearson’s correlations between components of vitamin D and androgen metabolism were all non-significant (Table 3). We also explored associations between markers of vitamin D status, androgens, and features of PCOS (Appendix A). Total and free 25(OH)D were negatively associated with HOMA-IR. Free 25(OH)D was also negatively associated with body fat percentage and hs-CRP. DBP was positively associated with hs-CRP. There were no associations between androgens and features of PCOS.

Multivariable linear regression between components of vitamin D and androgen metabolism is presented in Appendix A. Total 25(OH)D was positively associated with SHBG after adjusting for age and body fat percentage, however, this association became non-significant after further adjusting for HOMA-IR. There were no associations between components of vitamin D and androgen metabolism after adjusting for age, body fat percentage, HOMA-IR, and hs-CRP (Table 4).

## 4. Discussion

This cross-sectional study of pre-menopausal women with PCOS suggests that DBP is positively associated with total 25(OH)D, and negatively associated with free 25(OH)D. There were no associations between different markers of vitamin D status and testosterone, SHBG, or FAI, before and after adjustment for age, body fat percentage, HOMA-IR, and hs-CRP.

We have shown no associations between different markers of vitamin D status and androgens. This is consistent with other observational studies that have also found no associations between markers of vitamin D status and androgens [23,24]. Interestingly, most studies that have demonstrated associations between vitamin D and testosterone used immunoassays to quantify testosterone concentrations [7,8]. Immunoassays have numerous pitfalls, especially when measuring testosterone in women [25]. This is due to the limited ability of immunoassays to accurately quantify low levels of testosterone, cross-reactivity with steroid metabolites, precursors and conjugates, and regular changes in proprietary assay reagents [26]. Additionally, some of these studies have also failed to adjust for important confounders such as insulin resistance and/or BMI [7,8,27], which are often elevated in an individual’s PCOS, and directly affect androgens [28,29]. However, we saw no correlations between vitamin D components and androgens even before adjustment for these confounders. Nevertheless, a vitamin D–androgen link in PCOS might be explained through a number of pathways.

Vitamin D appears to influence aromatase activity in the ovaries, which affects testosterone concentrations. Kinuta et al. reported that mice with vitamin D receptor (VDR) knockouts have lower ovarian aromatase activity compared to wildtype counterparts, which was explained by decreased CYP19 gene expression, a key regulator of aromatase [30]. Vitamin D might also indirectly affect aromatase activity. Women with PCOS have elevated anti-Mullerian hormone (AMH) concentrations, which has been attributed to having a large number of follicles in the early stage of development, and greater AMH secretion per granulosa cell [31,32]. Increased AMH reduces aromatase activity in the ovary [33,34] and AMH appears to be positively associated with testosterone in women with PCOS [35]. Studies have reported inconsistent relationships between vitamin D and AMH in women with PCOS [36,37,38]. VDR polymorphisms may help explain these inconsistent findings; Szafarowska and colleagues recently reported that AMH was associated with VDR polymorphisms (Fok1 and Apa1), but not with 25(OH)D concentrations in women with PCOS [39]. Additional studies utilising gold-standard analytical techniques are required to confirm the link between vitamin D (including VDR polymorphisms), AMH, and androgens in women with PCOS.

There are number of limitations of this study. This study had a cross-sectional design, which limits comments on causation. We also had a relatively small sample of pre-menopausal women with PCOS and our findings need to be interpreted with caution. The CLIA assay we used does not differentiate between 25(OH)D2 and 25(OH)D3, and has cross-reactivity with other vitamin D metabolites such as 1,25(OH)2D2 and 1,25(OH)2D3 [40,41]. DBP gene polymorphisms were not studied and could have influenced binding affinities for 25(OH)D. Despite this, a previous study demonstrated similar prevalence of DBP gene polymorphisms in women with and without PCOS [42]. We also did not directly measure bioavailable and free 25(OH)D and instead, calculated these concentrations using previously validated formulae [21]. Future studies should include a larger sample size and measure relevant biochemical parameters using gold-standard liquid-chromatography–tandem mass spectrometry and VDR polymorphisms.

## 5. Conclusions

In conclusion, we found no associations between vitamin D metabolites, including free and total vitamin D and DBP and androgens in women with PCOS.

## Figures and Tables

**Table 1 nutrients-12-01219-t001:** Descriptive characteristics (*n* = 46).

	Mean ± SD	N
Age (years)	29.9 ± 5.6	46
Body Mass Index (kg/m^2^)	29.0 ± 5.5	46
Body Fat (%)	42.4 ± 9.2	44
Fasting Glucose (mmol/L)	4.7 ± 0.5	44
Fasting Insulin (mU/L)	17.4 (11.5, 24.9)	44
HOMA-IR	3.4 (2.1, 5.5)	44
Total 25(OH)D (nmol/L)	52.9 ± 18.9	44
Free 25(OH)D (pmol/L)	13.0 ± 5.4	39
Bioavailable 25(OH)D (nmol/L)	2.2 (1.4, 3.3)	41
Vitamin D-Binding Protein (ug/mL)	309.4 (219.6, 465.9)	42
Testosterone (nmol/L)	1.4 ± 0.7	46
SHBG (nmol/L)	44.4 ± 20.3	46
FAI	3.0 (2.0, 4.9)	46
Modified Ferriman-Gallwey Score	14.0 (11.0, 15.0)	46

Data presented as median (IQR) for non-normally distributed data. HOMA-IR: the homeostatic model assessment of insulin resistance. 25-hydroxyvitamin D: 25(OH)D; Sex Hormone-Binding Globulin: SHBG; Free Androgen Index: FAI.

**Table 2 nutrients-12-01219-t002:** Spearman’s correlations between markers of vitamin D status.

	Vitamin D-Binding Protein (ug/mL)
Total 25(OH)D (nmol/L)	**0.461 (0.003)**
Free 25(OH)D (pmol/L)	**−0.574 (<0.001)**
Bioavailable 25(OH)D (nmol/L)	−0.089 (0.602)

Correlation coefficient (*p*-value). 25-hydroxyvitamin D—25(OH)D. Bold values indicate statistical significance.

**Table 3 nutrients-12-01219-t003:** Pearson’s correlations between androgens and markers of vitamin D status.

	Total 25(OH)D (nmol/L)			Free 25(OH)D (pmol/L)	Bioavailable 25(OH)D (nmol/L) #	Vitamin D-Binding Protein (ug/mL) #
Testosterone (nmol/L)		0.044 (0.776)	−0.081 (0.622)		−0.094 (0.557)	0.080 (0.613)
SHBG (nmol/L)		0.261 (0.087)	0.195 (0.235)		0.041 (0.799)	0.004 (0.980)
FAI #		−0.094 (0.543)	−0.108 (0.511)		−0.090 (0.577)	0.087 (0.583)

Correlation coefficient (*p*-value); #: Spearman’s correlations. 25-hydroxyvitamin D: 25(OH)D; Sex Hormone-Binding Globulin: SHBG; Free Androgen Index: FAI.

**Table 4 nutrients-12-01219-t004:** Multivariable linear regression between androgens and markers of vitamin D status.

	Total 25(OH)D (nmol/L)	Free 25(OH)D (pmol/L)	Bioavailable 25(OH)D (nmol/L)	Vitamin D-Binding Protein (ug/mL)
Testosterone (nmol/L)	0.010 (−0.003, 0.024)	−0.028 (−0.086, 0.031)	0.121 (−0.062, 0.304)	0.001 (−0.0002, 0.003)
SHBG (nmol/L)	0.186 (−0.243, 0.615)	0.524 (−1.306, 2.354)	−0.436 (−6.242, 5.371)	−0.013 (−0.058, 0.032)
FAI	−0.001 (−0.008, 0.006)	−0.010 (−0.040, 0.019)	0.045 (−0.047, 0.138)	0.0004 (−0.0003, 0.001)

Values are unstandardized beta-coefficients (95% confidence intervals). 25-hydroxyvitamin D: 25(OH)D; Sex Hormone-Binding Globulin: SHBG; Free Androgen Index: FAI. Adjusted for age, body fat percentage, the homeostatic model assessment of insulin resistance (HOMA-IR) and hs-CRP. Free androgen index was log transformed.

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
