# Peer review of "The Relationship between Vitamin D Metabolites and Androgens in Women with Polycystic Ovary Syndrome"

_nutrients, 2020, doi:10.3390/nu12051219_

Round 1

Reviewer 1 Report

The manuscript is generally well-written, however I would like to point out some concerns about the layout: Table 1 looks superficially edited: decimal points should be used at all times, consistently and numbers should be cited in the text accurately. Moreover, the data presented clearly need some kind of visual representation to make it easier to understand.

The introduction part is somewhat inaccurate too. In line 48, there are four citations mentioned. Having read the articles cited as Nr. 7 and 9, neither one states significant association between androgen and vitamin D levels. In my opinion, these citations should be omitted. 

The sample size is really small, especially knowing what a common disease PCOS is. Having looked at the descriptive characteristics, I noticed that testosterone levels were really low compared to a PCOS population. According to the Endocrine Society Laboratory Reference Ranges, this sample is not even considered hyperandrogenic. The second line of the abstract of the manuscript states that 90% of PCOS women are hyperandrogenic. This is a clear inconsistency. If the patients of the sample had, however, hyperandrogenic symptoms, it should somehow be described, like adding the Ferriman-Gallwey scores.

Reviewer 2 Report

In this manuscript titled ‘The relationship between vitamin D metabolites and androgens in women with polycystic ovary syndrome’ the authors investigated the relationship between Vitamin D and androgen in PCOS women. They show that there were no associations between vitamin D metabolites and total testosterone or free androgen index after adjusting for all confounders. Further, there were no associations between vitamin D metabolites and androgens in women with PCOS. They also suggest that other studies that identified differences could be due to their unreliable techniques. This is an interesting manuscript even though, it shows negative data.

1.       The main issue this that the sample size for such a population based study is very small and the results needs to be interpreted with caution. Authors need to mention clearly that the study needs to be interpreted with caution.

2.       Did they exclude patients taking vitamin D or multivitamins based on self-reporting or based on medical records? They need to mention this in the manuscript.

3.       Please include the normal ranges of vitamin D for easy understanding by the readers.

Reviewer 3 Report

  1. In material and methods section, it would be necessary to list out information about every reagent, assay and platform used in the study.
  2. In Table 4 shown the model of adjusting age, body fat, HOMA-IR and CRP together, however, every confounder might have different effect directions to the outcome. Maybe it could be better to provide tables of adjusting age, body fat, HOMA-IR and CRP along in the supplementary data to clarify the effect of each confounders.

  3. The authors used rigorous and appropriate statistical methods in the study, and considered distribution of variables. Maybe larger sample size could help to give better results in normal distribution of bioavailable 25(OH)D, vitamin D Binding Protein and Free Androgen Index, for the data to more properly fit the linear regression model.
